# Identification and Characterization of Novel Sources of Resistance to Rust Caused by *Uromyces pisi* in *Pisum* spp.

**DOI:** 10.3390/plants11172268

**Published:** 2022-08-31

**Authors:** Salvador Osuna-Caballero, Nicolas Rispail, Eleonora Barilli, Diego Rubiales

**Affiliations:** Institute for Sustainable Agriculture, CSIC, Avda. Menéndez Pidal s/n, 14004 Córdoba, Spain

**Keywords:** phenotyping, plant breeding, *Pisum*, rust, quantitative resistance, *Uromyces pisi*, hypersensitive response

## Abstract

Pea rust is a major disease worldwide caused by *Uromyces pisi* in temperate climates. Only moderate levels of partial resistance against *U. pisi* have been identified so far in pea, urging for enlarging the levels of resistance available for breeding. Herein, we describe the responses to *U. pisi* of 320 *Pisum* spp. accessions, including cultivated pea and wild relatives, both under field and controlled conditions. Large variations for *U. pisi* infection response for most traits were observed between pea accessions under both field and controlled conditions, allowing the detection of genotypes with partial resistance. Simultaneous multi-trait indexes were applied to the datasets allowing the identification of partial resistance, particularly in accessions JI224, BGE004710, JI198, JI199, CGN10205, and CGN10206. Macroscopic observations were complemented with histological observations on the nine most resistant accessions and compared with three intermediates and three susceptible ones. This study confirmed that the reduced infection of resistant accessions was associated with smaller rust colonies due to a reduction in the number of haustoria and hyphal tips per colony. Additionally, a late acting hypersensitive response was identified for the first time in a pea accession (PI273209). These findings demonstrate that screening pea collections continues to be a necessary method in the search for complete resistance against *U. pisi*. In addition, the large phenotypic diversity contained in the studied collection will be useful for further association analysis and breeding perspectives.

## 1. Introduction

Pea (*Pisum sativum* L.) is a widely grown temperate grain legume. It is the second most cultivated legume in the world and the first in Europe, including both dry and green peas [1]. Its use extends to food and feed and represents a versatile and inexpensive protein source, bringing benefits to human health [2,3,4]. As for any other crop, pea production can be affected by a range of pests and diseases, among which pea rust has become a major concern worldwide [5].

Pea rust has been described to be incited by *Uromyces viciae-fabae* (Pers. de Bary) in tropical and subtropical regions [6] or by *U. pisi* (Pers.) (Wint.) in temperate areas [7,8]. *U. pisi* is a heteroecious macrocyclic fungus that completes its life cycle on *Euphorbia cyparissias* L. and *E. esula* L., which can grow in the vicinity of pea fields as spontaneous weeds and spread the fungal aeciospores over the crop [9]. When aeciospores infect pea, the uredial stage initiates a polycyclic infection, which results in the reduction in a photosynthetic area of an underdeveloped pod with yield losses up to 30% [10].

Chemical control with fungicides is effective to control rust, but it is expensive and has environmental side effects [11]. Alternative control methods, such as application of natural substances with fungistatic effects [12], the induction of systemic acquired resistance [13,14], or agronomic practices, such as intercropping [15,16], are being explored but are not yet available at a commercial level. Therefore, it is important to pay attention to the inherent resistance within *Pisum* spp. diversity because the development of resistant varieties through plant breeding remains the most economical and eco-friendly approach to control foliar diseases, such rust in pea [17]. Some efforts have been achieved in that direction to identify sources of resistance to *U. pisi* and to understand the genetic basis of resistance. Large pea collections have been screened under field and controlled conditions, but complete resistance has not been found so far [7,18]. The development of a RIL population with the partially resistant *P. fulvum* accession IFPI3260 as donor allowed the detection of four QTLs related to phenotypic disease variation [19,20].

The aim of this work was to expand the genetic background of *U. pisi* resistance in pea, evaluating new germplasm, including wild relatives, landraces, cultivars, and breeding lines from all over the world. The new resistant sources selected using a multi-trait index approach were characterized histologically to identify early resistance mechanisms. The components of resistance operating in each selected genotype are discussed according to their appropriate use in pea breeding programs for efficient and durable resistance to *U. pisi*.

## 2. Materials and Methods

### 2.1. Pisum ssp. Germplasm Origin

This study used a worldwide germplasm collection containing 320 pea accessions kindly provided by USDA (Department of Agriculture, Quezon City, Philippines), JIC (John Innes Centre, Norwich, UK), CRF-INIA (Centro Nacional de Recursos Fitogenéticos, Madrid, Spain), CGN (CPRO-DLO, Wageningen, The Netherland), IPK (Gatersleben, Germany), and ICARDA (International Centre for Agricultural Research in the Dry Areas, Beirut, Lebanon). The core collection represents the *Pisum* genera in taxonomy, geographic distribution, and phenotypic variation (Appendix A). All accessions were multiplied at the Institute for Sustainable Agriculture—CSIC at Cordoba, Spain, under field conditions before the experiments.

### 2.2. Pathogen Isolate and Multiplication

All experiments were performed with the *U. pisi* isolate Up-Co01 previously collected from naturally infested pea fields in Cordoba, Spain, and conserved at −80 °C. Before use, rust spores were multiplied on susceptible pea cv. Messire under controlled conditions (CC) to ensure availability of freshly collected spores at optimum conditions for inoculations, following the Sillero et al. [21] procedure with modifications. For this, two-week-old Messire plants were inoculated by dusting the plants with 1 mg urediospores per pot, mixed in pure talc (1:10, *v*:*v*), and incubated for 24 h at 20 °C in complete darkness and 100% relative humidity. Plants were then transferred to a growth chamber at 20 °C with a photoperiod of 14 h of light and 10 h of darkness and a light intensity of 148 µmol m^−2^ s^−1^. After 2 weeks, the fresh urediospores were collected using a vacuum spore collection device, dried, and stored until inoculation assays.

### 2.3. Field Experiments and Data Assessments

The pea collection was phenotyped over three crop seasons (2017/2018, 2018/2019, and 2019/2020) at Cordoba, Spain (Table 1), using the rust susceptible pea cv. Messire as control check, following an alpha lattice design with three replicates. To ensure optimal germination, pea seeds were first scarified and then sown in the field by early December each year, according to local practice. The experimental unit consisted of a single 1-m long row per accession with 10 seeds per row, separated from the adjacent row by 0.7 m.

Plants were inoculated at mid-March to ensure high and uniform levels of rust infection. The inoculation consisted of spraying plants with an aqueous urediospores suspension (±1.0 × 10^5^ urediospores mL^−1^) with Tween-20 (0.03%, *v*:*v*) as surface-active agent after sunset to benefit from the darkness and high relative humidity of the night. Disease severity (DS) was visually estimated as the percentage of canopy covered by rust pustules 30 days post inoculation (dpi) [7].

### 2.4. Controlled Condition Experiment and Assessments

Seeds of each accession were scarified and surface-sterilized for 20 min in a 20% solution of sodium hypochlorite and rinsed twice with sterile water for 20 min. Seed vernalization was induced for 3 days on wet tissue in a Petri dish at 4 °C in darkness and then shifted to 20 °C for 4 days for complete germination. Two germinated seeds per accession were sown in a 1:1 mixture of sand and peat per pot (35 × 35 cm) to finally leave one grown plant per pot for the evaluation. Each accession was replicated once. Pots were placed in a randomized complete block design and seedlings were inoculated when the third leaf was completely expanded (±12 days after sowing). Inoculation was carried out as described above for the CC multiplication of pea rust spores on cv. Messire. Then, plants were transferred to a growth chamber at 20 °C with a photoperiod of 14 h of light and 10 h of darkness and 148 µmol m^−2^ s^−1^ of irradiance at plant canopy level. The whole experiment was repeated three times leading to the evaluation of a total of six plants per accession.

Two days after inoculation, one leaflet of the third leaf was cut from each seedling and processed for histological assessments. The rest of the plants were maintained intact for macroscopical observations of rust development from 7 to 14 dpi by daily counting of the number of emerged pustules on a 1 cm^2^ marked area of the third leaf. These daily scorings of the first rust cycle were used to calculate the time when 50% of pustules were formed (latency period, LP50) and the monocyclic disease progress rate (MDPr) given by the slope of the regression line. To calculate MDPr, daily emerged pustules numbers were converted into relative pustule values expressed as the percentage of the most susceptible cultivar in the collection (cv. Erygel—JI1210). The last count was used to determine the final number of pustules cm^−2^ (infection frequency, IF). By 14 dpi, disease severity was also visually estimated as the percentage of canopy covered by rust, and infection type (IT) was assessed using the scale of Stakman et al. [22].

### 2.5. Histological Assessments

The leaflet samples collected at 2 dpi were bleached on filter paper dipped in fixative solution (absolute ethanol/glacial acetic acid, 1:1, *v*:*v*), and then they were boiled in 0.05% trypan blue in lactophenol/ethanol (1:2, *v*:*v*) for 10 min. Finally, they were cleared in a nearly saturated aqueous solution of chloral hydrate (5:2, *w*:*v*) to remove trypan blue from chloroplastic membranes as described in Sillero and Rubiales [23]. Histological observations were made using a phase contrast Leica DM LS microscope at ×400 magnification. Assessments were based on the observation of 25 random infection units per leaflet with three independent replicated leaflets per accession. For each infection unit, the number of hyphal tips and haustoria were counted allowing estimation of the early abortion rate (colonies without haustoria). Presence or absence of host cell necrosis associated to the infection units was also noted through the detection of autofluorescence upon cell excitation by UV light. The colony size, including perimeter and area, was also determined using a Levenhuk M1400 PLUS camera and LevenhukLite software.

### 2.6. Data Manipulation and Statistical Analysis

The control condition (CC) and field experiment datasets were analyzed separately. Control of data quality was performed individually for each trait through graphical inspection of residuals to assess normality, homogeneity of variance, and outliers’ detection. To ensure residuals normalization and variance stabilization, arcsine transformation was applied on the parameters expressed as percentages while square root transformation was performed for MDPr and IF and logarithmic transformation for LP_50_. A two-way ANOVA and post-hoc Tukey test were performed for post-haustorium histological studies.

For CC traits (MDPr, IF, DS_CC_, IT, and LP_50_), the experiment was analyzed using a linear mixed-effect model (LMM) according to the following equation:$$y_{ij}=\mu +\alpha_{i}+\tau_{j}+\varepsilon_{ij}$$
where *y_ij_* is the trait observed value for the *i*th genotype in the *j*th replicate (*i* = 1, 2, … 320; *j* = 1, 2, … 6); *α_i_* is the random effect of the *i*th genotype; *τ_j_* is the fixed effect of the *j*th replicate; and *ε_ij_* is the random error associated to *y_ij_*.

For field data where multi-environment trials were conducted, the linear model with interaction effect was used to analyze data:$$y_{ijk}=\mu +\alpha_{i}+\tau_{j}+{\left(\alpha \tau \right)}_{ij}+\gamma_{jk}+\varepsilon_{ijk}$$
where *y_ijk_* is the trait observed value in the *k*th block of the *i*th genotype in the *j*th environment (*i* = 1, 2, … 320; *j* = 1, 2, 3; *k* = 1, 2, 3); *μ* is the grand mean; *α_i_* is the effect of the *i*th genotype; *τ_j_* is the effect of the *j*th environment; *(**ατ)_ij_* is the interaction effect of the *i*th genotype with the *j*th environment; *γ_jk_* is the effect of the *k*th block within the *j*th environment; and *ε_ijk_* is the random error. In this case, the genotype effect and the interaction genotype × environment (G × E) effect were selected as random effects and environment, and replicates in environments were selected as fixed effects in the model.

In both models, the restricted maximum likelihood (REML) procedure was conducted to estimate the variance components of the linear mixed model to compute the predicted means (best linear unbiased prediction—BLUP) genotype values according to DeLacy et al. [24]. BLUPs were used as phenotypic data for subsequent correlations and genotype selection assessments. Variance components were also estimated in terms of coefficient of variation, following the formula:$$CV=\left(\sqrt{{\hat{\sigma }}^{2}}/\mu \right)\times 100$$
where *σ*^2^ is the variance and *μ* is the grand mean. The broad-sense heritability (*H*^2^) on an entry mean basis in all growing conditions was estimated following the Toker (2004) study [25] and calculated as:$$H^{2}=\frac{{\hat{\sigma }}_{g}^{2}}{{\hat{\sigma }}_{g}^{2}+{\hat{\sigma }}_{i}^{2}+{\hat{\sigma }}_{e}^{2}}$$
where *σ_g_*^2^ is the genotypic variance, *σ_i_*^2^ is the genotype-by-environment interaction variance, and *σ_e_*^2^ is the residual variance.

To select rust resistant accessions through traits evaluated under controlled conditions, multi-trait genotype-ideotype distance (MGIDI) [26], Smith-Hazel [27], and FAI-BLUP [28] indexes were performed. For field data, the Linn and Binns superiority measures were assessed using a non-parametric method for genotype selection [29]. Every index ranks the accession based on their rust resistance level, and the selected resistant genotypes were those accessions that appeared in the best positions over all four indices.

All data analyses were performed in R [30] with the “metan” [31] package for fitting LM/LMM interpretation and “ggplot2” [32] package for visualization.

## 3. Results

### 3.1. Phenotypic Response, Variance Components and Broad-Sense Heritability

All accessions showed rust symptoms under both controlled and field conditions albeit with variable intensities. A large phenotypic variation was observed in all trials for all traits, although accessions with moderate levels of disease symptoms were the most frequent in revealing a positive skewness, except for IT, as most accessions showed well-formed pustules (IT = 4; Figure 1g). The susceptible check cv. Messire showed a DS under field conditions of 28, 35, and 50 in 2018, 2019, and 2020, respectively, which was in all cases higher than the total mean of these environments (Figure 1).

Under CC, cv. Messire showed higher values than the total mean for MDPr, IF, IT, and DS CC and lower values for LP_50_. These values were far from the min and max value, showing a range from incomplete resistant to highly susceptible accessions. The lowest IT value was observed in accession PI273209, with moderate levels of macroscopically visible necrosis associated with rust pustules (IT = 2), whereas all other accessions displayed a fully compatible interaction (IT > 3, with IT = 4 being the most frequent).

The likelihood ratio test revealed significant difference between genotypes for all traits (*p* < 0.05). In CC, 63% of the phenotypic variance of DS was due to genetic differences between accessions. This trait also showed the highest broad-sense heritability (H^2^ = 0.86). By contrast, genotypic effect explained only 11.2% of the LP_50_ variance suggesting the low suitability of this parameter to predict the rust susceptibility/resistance response of a genotype. Under field conditions, the maximum genotypic effect was explained in 2020 with a 57% of the phenotypic variance and a broad-sense heritability of 0.80 (Figure 1). By contrast, the lowest genotypic effect was detected in 2018 explaining only 36% of the phenotypic variance and a broad-sense heritability of 0.63 (Figure 1).

These variations between environments are also affected by genotype x environment interactions. The minimum interaction coefficient was detected for 2019 with 11% and peaks in 2020 with 27% (Table 2). In most cases, the accuracy of the LMM applied in CC and field conditions is >75%, except for LP_50_.

### 3.2. Trait Correlations

Phenotypic correlations were calculated for each trait in each experimental condition. DS in the various field seasons were significantly correlated (*p* ≈ 0.5, *p* < 0.001) (Table 3). Adult plant responses in the field (DS in the various season) were significantly correlated with seedling responses under controlled conditions, with IF showing higher correlations, followed by MDPr, DS, and IT. On the contrary, LP_50_ in seedlings was poorly correlated with DS in the field, although it was correlated with DS in seedlings (*p* = −0.20, *p* < 0.01).

The strongest positive correlation was detected between MDPr and IF under CC (*p* = 0.96, *p* < 0.001) while the strongest negative correlation was detected between MDPr and LP50 (*p* = −0.21, *p* < 0.01) under controlled conditions.

### 3.3. Selected Rust Resistant Accessions

Grouping accession according to the species and subspecies they belong to did not reveal a group of accessions with higher level of resistance, due to the large variance within groups (data not shown). In addition, the interaction between these factors and the environment hampers direct selection of most resistant accessions (Appendix A).

Therefore, multi-trait indexes were applied to the dataset to drive selection of resistant accessions. This approximation points to several pea accessions that belong to different *Pisum* taxa and origin. Based on this multi-trait index selection of the nine most resistant accessions, the three most susceptible ones and three with intermediate levels were selected (Table 4).

### 3.4. Pea Resistance Mechanisms against U. pisi

To complement the macroscopic characterization of the pea panel in response to *U. pisi* inoculation, the selected accessions were further analyzed at histological level. The components of resistance of these selected accessions are presented in Table 5.

As expected, the most resistant accessions showed lower levels of infection with significantly lower values for all macro- and microscopical traits. No differences among accessions were observed at early stages of the infection (Figure 2a), with no significant differences for spore germination or appressoria formation. Post-appressoria infection events (Figure 2b), including infection unit area, infection unit perimeter, number of hyphal tips, and haustoria, are lower in resistant accessions than in susceptible ones (*p* < 0.05). By contrast, infection unit area, infection unit perimeter, and hyphal tips were not significantly different between moderately and highly susceptible accessions. However, there is a significant difference between moderately and highly susceptible accessions in terms of haustoria number/infection unit.

Significant variation was also observed between partially resistant and susceptible accessions in terms of early abortion (infection units forming at least in HMC in contact with a mesophyll cell but failing to form any haustoria) not associated with host cell necrosis. Such early abortion was particularly high (over 23%) in JI198, JI199, and IFPI3260, followed by JI224, CGN10205, CGN10206, and PI347372 (over 12%) (Table 5). No or negligible levels of host cell necrosis were observed by 48 hpi on any of these early aborted colonies. Host cell necrosis was also absent or low (<7%) in established colonies of all studied accessions, except PI273209, where it reached 20%, (Figure 3, Table 5); what was macroscopically visible as small pustules surrounded by necrotic halo (IT = 2) did not prevent sporulation but hampered it, resulting in incomplete resistance based on late acting hypersensitivity (Figure 4).

## 4. Discussion

Pea rust, caused by *U. pisi* and *U. viciae-fabae*, is a major disease responsible for serious yield losses worldwide [5]. *U. pisi* is the principal agent causing pea rust in temperate regions [33], while *U. viciae-fabae* is more widely distributed in tropical and subtropical areas [34]. Only moderate levels of resistance against *U. pisi* have been identified so far [7].

In this study, we macroscopically analyzed the response to rust caused by *U. pisi* in a worldwide *Pisum* spp. collection of 320 accessions, meticulously selected to ensure a wide range of phenotypic and genetic variation ranges. The variability found within the collection under field conditions was moderately correlated between seasons supporting the strong influence of the weather conditions and the genotype x environment interactions as suggested previously for pea rust [35,36]. Rust DS was higher in 2020, which was characterized by a climate more favorable to *U. pisi* development due to a higher mid temperature and higher relative humidity [37] favoring disease infection, in agreement with previous studies on *U. viciae-fabae* [38,39]. Infection in 2018 were weakly correlated with traits measured under controlled conditions (CC) while field data recorded in 2019 and 2020 seasons were characterized by more favorable climates, which were better correlated with these traits. Correlation in adult plants and seedlings in response to rust disease can be affected not only by temperature and relative humidity but also by the polycyclic effect over the host and leaf age, which can express or not express the genes behind the resistance mechanism. Effect of temperature and leaf age has been well studied [40] in other pea foliar disease, such as powdery mildew, but it is still unknown for rust.

Variability found in this collection according to disease severity of seedlings in CC to rust was higher than other studies performed in lathyrus [41] and vetch [42] against *U. pisi* and comparable to pea studies in the same pathosystem where check cultivar Messire showed similar DS (~50%) [7]. The screening techniques are well established and the *U. pisi*–legume pathosystem response confirmed the reproducibility of the method described by Sillero et al. (2006). All traits assessed under controlled conditions showed moderate to high positive correlation between them, except LP_50_, which showed a negative one, in agreement with previous studies [7]. Latency period of *U. pisi* development increases with the quantitative resistance level of pea genotypes similarly to other biotrophic pathogens [43]. Accordingly, it was negatively correlated with MDPr, IT, and DS. However, final IF was not correlated with the latency period in our collection, suggesting that the final number of pustules emerging on a leaf is an independent event controlled by mechanisms distinct from those controlling the incubation period and the pre-sporulation symptoms involved in the latency period [44]. In addition, latency period in plant diseases is very sensitive to variations in disease expressions, including those due to phenotypic plasticity [45], defined as the ability of individual genotypes to produce different phenotypes when exposed to different environmental conditions [46]. A study carried out in wheat rust caused by *Puccinia triticina* revealed that, even in a clonal lineage population, significant differences were present in the latency period within identical pathotypes [47].

In this study, resistance based on DS reduction without host cell necrosis was the most common response of the collection under field and controlled conditions for adult plants and seedlings, respectively. This incomplete resistance not associated with hypersensitive response is known as partial resistance. It is characterized by a decrease in IF and DS while IT is high [48], which means that plants harbor a lower number of pustules but those that do form develop normally. These components of quantitative resistance have demonstrated they are more durable than major gene resistance on average [49], so they are considered desirable traits for an effective field rust resistance, also in other crops, such as cereals [50,51,52]. The partial resistance reported here has been observed previously in pea against both *U. viciae-fabae* [53] and *U. pisi* [18]. We support other studies in the identification of PI347321 and IFPI3260 accessions as partially resistant [7]. In addition, we have identified novel sources of partial resistance in the accessions JI224, BGE004710, JI198, JI199, CGN10205, and CGN10206. The level of partial resistance of two of these accessions, JI224 and JI198, was similar to the highest level of resistance previously detected (DS~10%) [7]. The additional resistant accessions, BGE004710, JI 199, CGN10205, and CGN10206, had a slightly higher level of partial resistant with DS < 7%. Those results expand the genetic source of resistance by providing new accessions with levels of partial resistance similar to the previously described IFPI3260, which was to date the most resistant accession available against *U. pisi*.

On the one hand, there is only one *P. sativum* subsp. *sativum* accession described with high partial resistance levels, BGE004710. Its origin is assigned to Mogadouro in Portugal, according to its passport, where the incidence of rust caused by *U. pisi* and *U viciae-fabae* is elevated [54]. Due to this taxonomy, its potential use to transfer the minor genes conferring the partial resistance to pea cultivars in temperate climates is high. On the other hand, it is also possible to use other wild relatives and landraces described here as donor of partial resistance, since crosses between wild pea species and subsp. *sativum* cultivars have been already explored [55]. In fact, it is known that wild pea relatives from *P. fulvum* work as a donor of resistance to biotic stresses, such as insect, diseases, or weeds [56]. *P. sativum* subsp. *elatius* var. *elatius* has been used for breeding purposes increasing the nutritional value of peas, without being exploited as a source of resistance to biotic stresses so far [57]. In this context, the additional partial resistance sources detected here belonging to *P. sativum* subsp. *sativum* and *P. sativum* subsp. *elatius* var. *elatius* could allow the localization of new genome regions associated with rust resistance in pea, in addition to those currently described in the *P. fulvum* [20]. Recently, a panel of *Lathyrus sativus* inoculated with *U. pisi* revealed novel loci behind partial resistance mechanisms using an association mapping approach [58]. Similarly, the present pea panel would be valuable to expand the genetic bases of resistance for future breeding of rust resistant pea.

To integrate all traits scored in controlled and field conditions a multi-trait index approach was applied to differentiate accessions. Since the first index was proposed by Smith [27], multi-trait selection indices are established strategies to select superior genotypes in plant breeding and provide the breeder with an objective rule for evaluating and selecting several traits simultaneously [59]. However, the Smith-Hazel index can be affected by multicollinearity problems, providing erroneous conclusions and inefficient conservation measures [60]. To offset this possibility, we applied in parallel two additional multi-trait indices, MGIDI and FAI-BLUP, that have been established for plant breeding selection and are free from weighting coefficients and multicollinearity issues [24,26]. In addition, a non-parametric index that includes the genotype × environment interaction was used to select the most resistant pea genotypes to *U. pisi* in the field [29]. The results obtained from these indexes are quite similar in the cases of FAI-BLUP and MGDI, while SH and LIN provide different rankings of the superior genotypes. Even if one of the index criteria performed an erroneous selection, this was compensated by the other three indices supporting the usefulness of applying simultaneous indices. Based on this methodology the nine best-performing pea accessions in the four indices were selected and used to assay the underlying resistance mechanisms to *U. pisi* infection histologically.

In the microscopical study, the variation in urediospores germination and appressoria formation over stoma did not affect rust severity, suggesting that the resistance mechanisms took place after formation of substomatal vesicles [15,61]. One of the most efficient post-appressoria resistance mechanisms is the early abortion of colonies that failed to form any haustoria in mesophyll cells. Here, we detected a high proportion of early colony abortion in the resistant accessions JI198, JI199, and IFPI3260. Histological and biochemical studies in pea rust suggested early abortive colonies are explained by a physical barrier to successful infection due to a lignification process in the mesophyll cells around the infection unit [13,62]. Additional assessments are currently underway to confirm the involvement of host cell wall strengthening in these accessions. When this first mechanism failed and the first haustoria mother cells develop a haustorium into mesophyll cell, a second resistance mechanism may impede penetration of secondary hyphae, reducing the number of haustoria per infection unit and, therefore, decreasing the colony size and the number of hyphal tips. All selected accessions show some degree of this penetration resistance, which was particularly visible for IFPI3260, BGE004710, and PI347321, showing the smallest infection unit size and hyphae number. A similar resistance mechanism had been found in pea–rust studies against *U. viciae-fabae* [53] and *U. pisi* [18] and in other legume–rust pathosystems [48,63,64].

Hypersensitive response (HR) is another post-appressorium formation mechanism associated with disease resistance. HR has been described previously against *U. viciae-fabae* in some legume crops, such lentil [65,66] and faba bean [23,67], although it is not the most common resistance mechanism against rust [23,68]. Here, low levels of hypersensitive reaction to *U. pisi* infection were observed in accessions JI224, BGE004710, and JI199 showing less than 7% of infection units associated with cell death but revealing a compatible reaction with high infection type (IT = 3++, 3 and 3, respectively). On the contrary, PI273209 that displayed a considerable percentage of necrotic mesophyll cells (20%) showed an incomplete reaction associated with macroscopically visible necrosis (IT = 2). In addition, this accession also showed non-hypersensitive resistance supported by histological results, reducing hyphal growth and hampering haustorium formation resulting in reduced disease severity despite some well-formed pustules. These observations indicate that a combination of both hypersensitive and non-hypersensitive resistance operates in PI273209 against *U. pisi*, but that the early abortion of colonies is not associated with cell death. This mechanism reveals that non-hypersensitive reaction can occur before and after haustoria formation, but HR only takes place after haustoria formation. This type of incomplete resistance has been described as “late acting” hypersensitive rust resistance, and it allows some haustoria failing to form due to hypersensitive cell death but others forming successfully [69]. Incomplete resistance associated with HR has been reported before in pea against *U. pisi* or *U. viciae-fabae*. The use of fluorescence microscopy and digital image technology was particularly useful to study these resistance components, allowing the detection of necrotic host cells and precisely measuring the colony area and perimeter similar to Rubiales and Sillero [23]. In other legume–rust pathosystems, this HR type has been described as monogenic, allowing the identification of genes *Uvf-1*, *Uvf-2*, and *Uvf-3* conferring hypersensitive resistance against *U. viciae-fabae* in faba bean [70,71] and gene *Rpp2* conferring hypersensitive resistance against *Phakopsora pachyrihizi* in soybean [72]. Consequently, future studies are needed to determine the genetic inheritance behind this resistance mechanism that will complement the actual genetic basis conferred by QTL’s UpDSII, UpDSIV, and UpDSIV.2, responsible for the genetic variance in partial resistance caused by *U. pisi* in pea wild relatives [20].

In conclusion, this study allowed the identification of new resistance sources from a wide collection of pea accessions and confirmed the importance of crop core collection to identify traits of interest. In addition to identifying additional sources of partial resistance with a similar level to the highest previously described resistant accessions, we identified a moderate level of late-acting HR in one of our accessions, which had never been described before in the pea—*U. pisi* pathosystem. Including this accession, together with the additional sources of partial resistant in our breeding programs, should broaden the genetic bases of resistance, which is key for a more durable resistance. In addition, these novel resistance sources can be the base for further studies to establish the genetic, biochemical, and molecular nature of rust resistance in pea.

## Figures and Tables

**Figure 1 plants-11-02268-f001:**
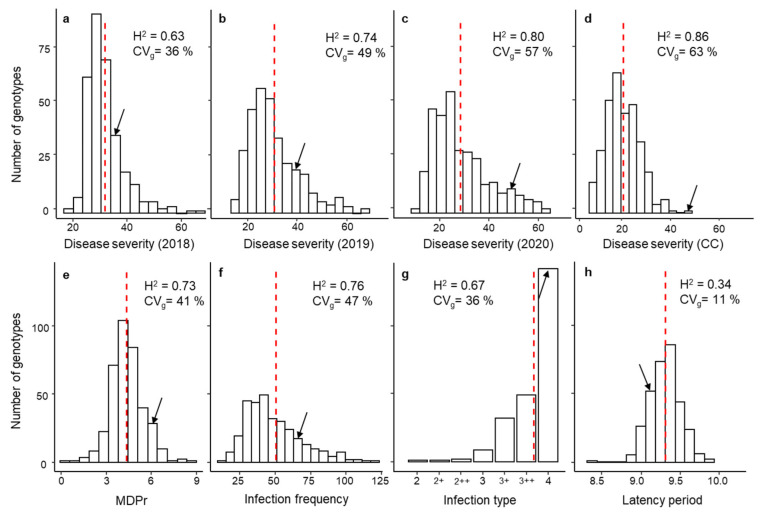
Phenotypic variation in rust response adjusted means (BLUPs) among 320 pea accessions after infection with *U. pisi*. (**a**–**d**) show the DS plants under field conditions during 2018, 2019, and 2020 seasons and DS seedlings under controlled conditions (CC), respectively. (**e**–**h**) show seedlings under CC, the monocyclic disease progress rate (MDPr), infection frequency (IF), infection type (IT), and latency period (LP50), respectively. Genotypic coefficient of variation (CV_g_) and heritability (H^2^) are also shown. Dashed red lines and black arrows indicate the overall mean and where the susceptible control Messire is located, respectively.

**Figure 2 plants-11-02268-f002:**
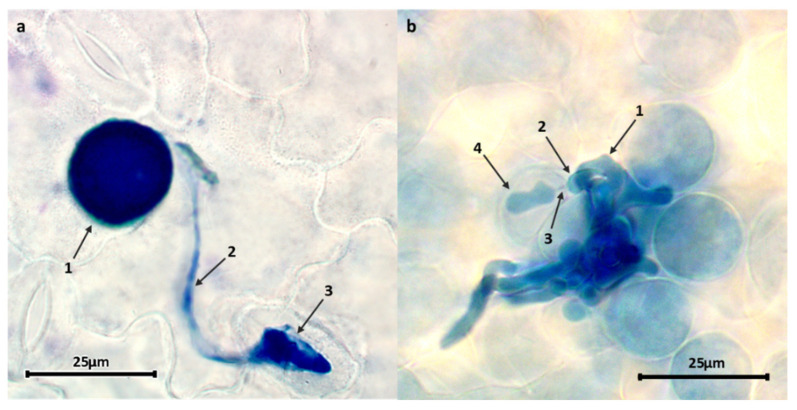
*U. pisi* structures infecting pea cells at 48 hpi. (**a**) Early rust infection event: a germinated urediospore (1) develops a germ tube (2) that differentiates an appressorium over a stoma (3). (**b**) A colony with hyphae growing between mesophyll cells (1) differentiates an haustorial mother cell (HMC) (2) which invaginates into the mesophyll cell via a neckband (3) forming an haustorium (4).

**Figure 3 plants-11-02268-f003:**
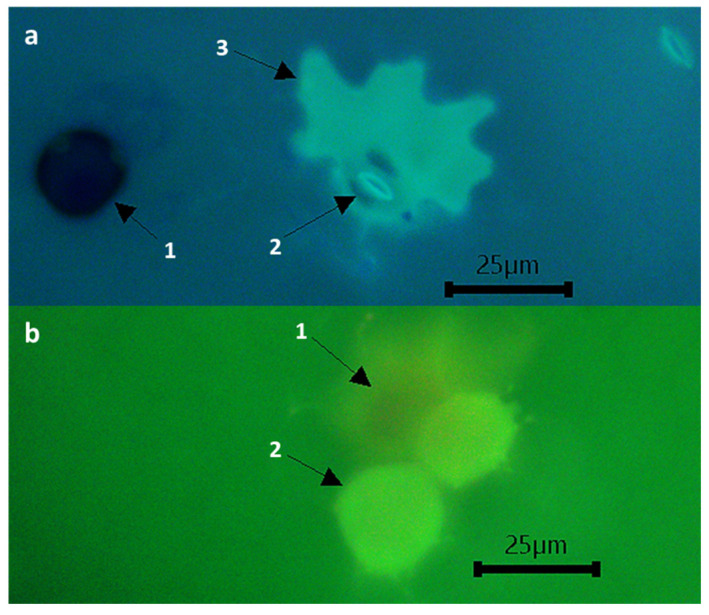
Different cells attacked by *U. pisi* colonies in PI273209 accession: (**a**) Epidermic tissue visualized with fluorescent blue filter at 48 h post inoculation showing a germinated urediospore (1), the stoma (2), and a dying epidermal cell (3); (**b**) Mesophyll tissue visualized with fluorescent green filter showing the substomatal space (1) and two dying mesophyll cells (2).

**Figure 4 plants-11-02268-f004:**
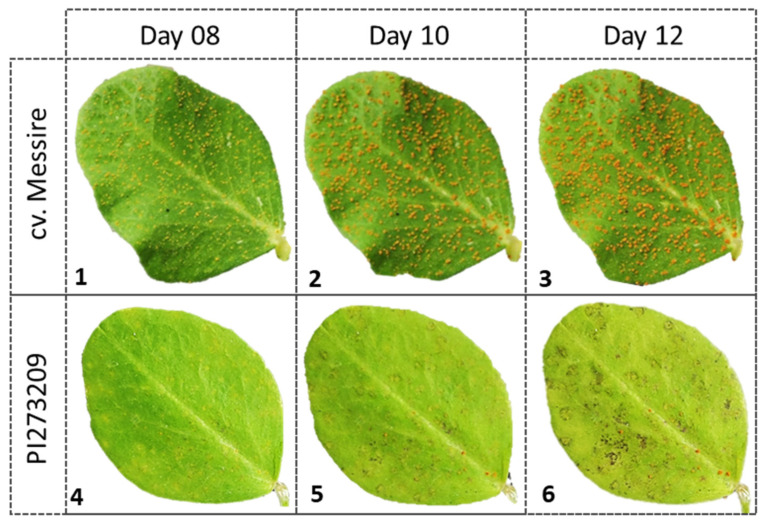
Rust symptoms progression in cv. Messire and PI273209 accession leaves. (**1**–**3**) show the macroscopic rust progression in cv. Messire at 8, 10, and 12 dpi, respectively. (**4**–**6**) show the macroscopic rust progression in PI273209 accession at 8, 10, and 12 dpi, respectively.

**Table 1 plants-11-02268-t001:** Description of the environments of the trials for the multi-environment study during the crop cycle from December to May.

Environments	Season	Site (Decimal Degrees Coordinates)	Soil Type	Soil pH	Organic Matter (g/110 g)	Available Phosphorus (mg/kg)	C:N Ratio	Average T_max_ (°C)	Average T_min_ (°C)	Average RH (%) *	Rain (mm)
Co-18	2017–2018	37.862875, −4.791796	Vertisol	-	-	-	-	18.37	6.17	75.2	472
Co-19	2018–2019	37.864470, −4.789733	Vertisol	7.8	0.7	9.9	7.25	21.04	5.78	63.4	127
Co-20	2019–2020	37.866372, −4.787661	Vertisol	-	-	-	-	21.00	8.45	73.9	382

* Relative humidity data taken from *U. pisi* inoculation to disease severity assessment.

**Table 2 plants-11-02268-t002:** Statistics performed for disease components studied in CC and under field conditions. Arithmetic mean ± standard error (SE), minimum and maximum values, skewness, accuracy of the selection in LMM applied, and their likelihood ratio test for genotype effect (LRT). The percentage of genotype-by-environment interaction coefficient of variation (CV_i_) is also shown with a significance *p* < 0.01.

	Trait	Mean ± SE	Skewness	Min	Max	Accuracy	LRT	CV_i_
Controlled Conditions	MDPr	5.7 ± 0.01	0.28	2.66	8.75	0.87	280	-
	IF	50.3 ± 1.19	2.33	1.00	387.00	0.87	264	-
	IT	3.8 ± 0.01	−2.05	2.00	4.00	0.82	180	-
	LP_50_	9.3 ± 0.02	−0.02	6.71	12.56	0.58	11	-
	DS_CC_	20.1 ± 0.29	0.53	2.00	60.00	0.93	549	-
Field Season	DS_2018_	27.5 ± 0.49	1.73	2.00	60.00	0.79	111	27
	DS_2019_	26.2 ± 0.56	1.06	1.00	65.00	0.86	205	11
	DS_2020_	28.8 ± 0.59	0.95	1.50	65.00	0.89	299	17

**Table 3 plants-11-02268-t003:** Pearson (*p*) correlation coefficient between traits evaluated under field and controlled conditions and calculated from adjusted mean (BLUPs) values from the 320 pea accessions.

	Field Conditions			Controlled Conditions			
	DS_2018_	DS_2019_	DS_2020_	MDPr	IF	LP_50_	DS_CC_
**Field conditions**							
DS_2019_	0.46 ***						
DS_2020_	0.46 ***	0.55 ***					
**Controlled conditions**							
MDPr	0.21 ***	0.44 ***	0.39 ***				
IF	0.21 ***	0.46 ***	0.40 ***	0.96 ***			
LP_50_	−0.04	0.01	−0.04	−0.21 **	−0.05		
DS	0.14 *	0.28 ***	0.30 ***	0.58 ***	0.57 ***	−0.20 **	
IT	0.18 *	0.29 ***	0.25 ***	0.45 ***	0.42 ***	−0.20 **	0.33 ***

* *p* < 0.05, ** *p* < 0.01, *** *p* < 0.001.

**Table 4 plants-11-02268-t004:** Selected accessions representing three rust response levels (partial resistant, intermediate, and highly susceptible) through multi-trait index approach (FAI-BLUP, MGIDI, Smith-Hazel (SH), and Lin-Bin (LN), respectively).

Rust Level	Bank Code	Species	Origin	DS_field_ (%)	DS_CC_ (%)	Ranking Indices (1–320)			
						FAI-BLUP	MGIDI	SH	LIN
Resistant	CGN10206	*P. sativum* subsp. *elatius*	Unknown	3.0	5.2	1	1	1	4
	CGN10205	*P. sativum* subsp. *elatius*	Turkey	6.0	5.5	2	2	2	1
	PI273209	*P. sativum* subsp. *elatius*	Russia	3.3	3.0	3	3	3	2
	IFPI3260	*P. fulvum*	Syria	4.3	6.7	4	4	6	3
	BGE004710	*P. sativum* subsp. *sativum*	Portugal	7.7	6.5	7	8	7	10
	JI199	*P. sativum* subsp. *elatius*	Israel	5.0	6.5	5	5	4	5
	JI198	*P. sativum* subsp. *elatius*	Israel	5.8	11.6	6	6	8	6
	PI347321	*P. sativum*	India	11.0	4.5	9	9	10	11
	JI224	*P. fulvum*	Israel	4.3	10.0	10	10	12	9
Intermediate	PI347372	*P. sativum*	India	16.3	16.1	155	155	162	150
	PI143483	*P. sativum*	Azerbaijan	17.0	22.1	160	160	167	156
	PI324705	*P. sativum*	France	15.7	21.0	175	175	184	169
Susceptible	PI162910	*P. sativum*	Paraguay	34.5	27.6	318	318	318	320
	PI204667	*P. sativum* subsp. *sativum*	Netherland	30.0	45.8	319	319	319	318
	JI1210	*P. sativum* subsp. *sativum*	France	40.7	28.6	320	320	320	315

**Table 5 plants-11-02268-t005:** Microscopic evaluation of *U. pisi* colonies 48 h post-inoculation in the selected genotypes. Values, per column, followed by different letters differ significantly at *p* < 0.05.

Bank Code	DS_CC_ (%)	IT	Infection Unit Area (µm²)	Infection Unit Perimeter (µm)	No. Hyphal Tips/Infection Unit	Early Abortion (%)	No. Haustoria/Established Colony	Established Colonies Associated with Host Cell Necrosis (%)
PI347321	4.5 ^d^	3++	490.8 ^c^	184.0 ^d^	3.3 ^c^	8.3 ^c^	1.9 ^ab^	0 ^d^
JI224	10.0 ^bc^	3++	792.5 ^b^	302.3 ^b^	5.8 ^ab^	12.5 ^b^	1.7 ^a^	2.5 ^c^
BGE004710	6.5 ^c^	3	592.6 ^bc^	211.6 ^cd^	3.6 ^c^	10.6 ^c^	1.6 ^a^	4.3 ^bc^
PI273209	3.0 ^d^	2	784.3 ^b^	277.5 ^b^	5.4 ^b^	10.0 ^c^	1.9 ^ab^	20.0 ^a^
JI198	11.6 ^bc^	3	941.2 ^ab^	374.3 ^a^	6.8 ^a^	23.7 ^a^	1.8 ^ab^	0 ^d^
JI199	6.5 ^c^	3	765.0 ^b^	270.3 ^bc^	5.0 ^b^	23.3 ^a^	1.6 ^a^	6.7 ^b^
CGN10205	5.5 ^cd^	3	545.4 ^c^	195.7 ^d^	3.8 ^c^	13.3 ^b^	1.6 ^a^	0 ^d^
CGN10206	5.2 ^cd^	3	606.4 ^bc^	240.2 ^c^	4.8 ^bc^	13.3 ^b^	1.5 ^a^	0 ^d^
IFPI 3260	6.7 ^c^	4	497.5 ^c^	190.8 ^d^	3.7 ^c^	23.3 ^a^	1.7 ^a^	0 ^d^
PI143483	22.1 ^ab^	4	1170.7 ^a^	360.5 ^ab^	6.1 ^ab^	0 ^d^	2.3 ^b^	0 ^d^
PI347372	16.1 ^b^	4	1074.9 ^a^	356.5 ^ab^	7.7 ^a^	12.9 ^b^	2.3 ^b^	0 ^d^
PI324705	21.0 ^ab^	4	662.6 ^b^	222.6 ^c^	3.8 ^c^	0 ^d^	2.3 ^b^	0 ^d^
PI162910	27.6 ^ab^	4	1202.0 ^a^	395.7 ^a^	7.9 ^a^	0 ^d^	3.4 ^c^	0 ^d^
PI204667	45.8 ^a^	4	1024.8 ^a^	334.0 ^b^	6.2 ^ab^	0 ^d^	3.5 ^c^	0 ^d^
JI1210	28.6 ^ab^	4	1006.1 ^a^	292.0 ^b^	5.9 ^ab^	0 ^d^	3.2 ^c^	0 ^d^

## Data Availability

Publicly available datasets were generated and analyzed in this study. This data can be found here: https://github.com/SalvaOsuna/Rust-collection.git (accessed on 4 May 2022).

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
