# Peer review of "Identification and Characterization of Novel Sources of Resistance to Rust Caused by Uromyces pisi in Pisum spp."

_plants, 2022, doi:10.3390/plants11172268_

Round 1

Reviewer 1 Report

Congratulations, a very thorough study which can be greatly improved with more in depth interpretation.

For effective field resistance in which low levels of rust infection are of minor consequence, the demonstrated partial resistance in both elatius and fulvum and one sativum, is possibly sufficient, durable and sustainable. Very low levels of infection and production of rust spores reduces the risk of evolution of more aggressive rust strains in the presence of immunity (e.g. with 100% abortion), and rust strains from partial resistance may be more competitive than those capable of overcoming immunity. Discussion could explore the likely sustainability of partial resistance in the field ( see Buddenhagen, sick plots for GM cotton in Australia).

An additional treatment could have been harvest of mixed strains of rust spores from infection of different pea varieties, for comparison of resistances expressed with infection of only one rust strain. Partial resistance to multi-strain rust infection would add confidence that high partial resistance was both durable and sustainable for long term for adequate field resistance.

Discussion could explore the passport provenance of the one sativum accession with high partial resistance. Was it from a locality very favourable to rust development? and was it genetically different from that expresssed in elatius? and how diverse  were the genetic expressions in elatius and fulvum? Could the partial resistances be stacked, although probably not needed at present.

A brief discussion could examine the feasibility of transferring partial resistance from elatius/fulvum to sativum. and need for further study on the possible novelty of these sources from wild relatives. The importance of wild relatives for accessing novel sources of resistance could be usefully examined. This seems to be a reason why relatives were included in this study, therefore it should be discussed in this study.

Author Response

Response to Reviewer 1 Comments

Reviewer: Congratulations, a very thorough study which can be greatly improved with more in depth interpretation.

Authors: Authors would like to thank the reviewer 1. We appreciate the reviewer insightful suggestion. All the comments to improve the manuscript (attached) were considered useful and were incorporated, as we describe below in red color:

Reviewer: For effective field resistance in which low levels of rust infection are of minor consequence, the demonstrated partial resistance in both elatius and fulvum and one sativum, is possibly sufficient, durable and sustainable. Very low levels of infection and production of rust spores reduces the risk of evolution of more aggressive rust strains in the presence of immunity (e.g. with 100% abortion), and rust strains from partial resistance may be more competitive than those capable of overcoming immunity. Discussion could explore the likely sustainability of partial resistance in the field ( see Buddenhagen, sick plots for GM cotton in Australia).

Authors: Authors think that this is an excellent suggestion. We have complete the paragraph related to partial resistance in the discussion highlighting its relevance as durable field rust resistance in pea (lines 339 – 343). In addition, references were added as relevant source of information to support your suggestion.

Reviewer: An additional treatment could have been harvest of mixed strains of rust spores from infection of different pea varieties, for comparison of resistances expressed with infection of only one rust strain. Partial resistance to multi-strain rust infection would add confidence that high partial resistance was both durable and sustainable for long term for adequate field resistance.

Authors: Thank you for pointing this out. Although we agree that this is an important consideration, it is cannot be analyzed in this manuscript because the magnitud of experimental units sown in field (n = 1083). The artificially inoculations of every rust strains to every plot will consume too much time and efforts to be abordable. For sure, we will consider this for coming studies because now we have selected the most appropiated accessions to show this purpose.

Reviewer: Discussion could explore the passport provenance of the one sativum accession with high partial resistance. Was it from a locality very favourable to rust development? and was it genetically different from that expresssed in elatius? and how diverse were the genetic expressions in elatius and fulvum? Could the partial resistances be stacked, although probably not needed at present.

Authors: Lines 354 - 358 have been updated, such that include the discussion about BGE004710 accession. Unfortunately, we do not have genomic information about accessions, so we appreciate the reviewer insightful suggestion and agree that it would be useful to demostrate the differences behind resistance origin. However that analysis is beyond the scope of our manuscript, wich aims only to show novel sources of resistance found within the collection studied and discuss the new mechanisms involved.

A brief discussion could examine the feasibility of transferring partial resistance from elatius/fulvum to sativum. and need for further study on the possible novelty of these sources from wild relatives. The importance of wild relatives for accessing novel sources of resistance could be usefully examined. This seems to be a reason why relatives were included in this study, therefore it should be discussed in this study.

Authors: We think the suggestion is already covered in the manuscript, between lines 359 - 364. However, we have improved the information provided about transferring partial resistance from elatius/fulvum to sativum with the available bibliography in lines 358 - 361.

Reviewer 2 Report

My comments and questions are included as comment boxes in the attached pdf. Most suggestions were to improve English language aspects. I would suggest that the authors have their manuscript critically appraised by someone proficient in English scientific writing.

Author Response

Response to Reviewer 2 Comments

Reviewer: My comments and questions are included as comment boxes in the attached pdf. Most suggestions were to improve English language aspects. I would suggest that the authors have their manuscript critically appraised by someone proficient in English scientific writing.

Authors: Authors appreciate sincerely the reviewer’s comments and advices. The typography and grammatical errors have been corrected on the manuscript (attached) accordingly to the suggestions received. In addition, the manuscript have been deeply revised in order to improve the English language and style.

All the comments related to clarify the manuscrpit were considered useful and were incorporated, as follows:

  • Line 99: Disease severity (DS) scale clarification. The visual estimation of the canopy covered by rust pustules does not follow a fix scale, it is a subjective calculation. However, we appreciate the comment and we have added a reference where the same methology to determinate DS in field was described.
  • Line 174: Reference added to broad-sense heritability.
  • Line 200: The reviewer is right. Adult plants instead that seedlings under field conditions.
  • Line 222: Table 2 title clarified
  • Line 298: Figure 4 definition clarified
  • Line 339: The definition and a reference about phenotypic plasticity have been added.
